# Quality of life in Arab children with congenital heart disease

**Latefa Ali Dardas** [1]*, **Wei Pan**[2], **Ahmad Imad Hamdan**[3], **Raghed Abdel Hay Abu Jabeh**[4], **Ahmad Eid Ashakhanba**[5], **Omar Sami Abdelhai**[6], **Mohammad Naim Abid**[7], **Hashim Ahmad Mohammad**[8], **Iyad Al-Ammouri**[9]

1 School of Nursing, The University of Jordan, Amman, Jordan, 2 Duke University School of Nursing, Durham, North Carolina, United States of America, 3 Al Bashir Hospital, Amman, Jordan, 4 Shmaisani Hospital, Amman, Jordan, 5 Institute For Family Health, Clinical Psychologist, Amman, Jordan, 6 Al-Khalidi Medical Center, Amman, Jordan, 7 Marka Specialty Hospital, Amman, Jordan, 8 Internal Medicine Resident, Hamad Medical Corporation, Doha, Qatar, 9 Pediatric Cardiology School of Medicine, The University of Jordan, Amman, Jordan

* l.dardas@ju.edu.jo

## Abstract

### Background and purpose

Management strategies for children with congenital health diseases (CHDs) should encompass more than just the medical aspect of the disease and consider how heart diseases affect their everyday activities and, subsequently, their quality of life (QoL). Global studies witnessed a greater emphasis on studying the QoL associated with CHD. However, there is still a great lag in such data in the Arab region. The purpose of this study was to evaluate QoL in children with CHD using an Arab sample from Jordan. The specific objectives were twofold: (1) to contrast the assessments of children's QoL reported by their parents with those reported by the children themselves, and (2) to assess the factors that influence the QoL of children with CHD.

### Methods

A total of 79 children aged 2–18 with a confirmed diagnosis of CHD were included in the study, along with their mothers. Of them, 38.0% were girls, 67.1% were diagnosed with non-cyanotic CHD, 58.2% had a severe CHD, 92.4% had undergone at least one operation, 81.0% had repaired defects, 13.9% underwent palliated procedures, and 24.1% were admitted to a neonatal intensive care unit after delivery. The Pediatric Quality of Life Inventory was used to assess QoL of children with CHD. Both children's and parents' reports of QoL were analyzed using paired-sample *t*-tests, ANOVAs, and multiple linear regression.

### Results

Older children reported significantly lower QoL scores, whereas there were no differences in parents-reported QoL scores across different children age groups. There was a divergence in perceptions of QoL between parents-reported and children-reported scores with parents reporting significantly lower scores. The children-reported QoL in this study seemed to be

**Data Availability Statement:** The data underlying the results presented in the study are available from OSF: https://osf.io/qs9m7

**Funding:** LA received financial funding from the University of Jordan Deanship of Scientific Research. The funder had no role in study design, data collection and analysis, decision to publish, or preparation of the manuscript.

**Competing interests:** The authors have declared that no competing interests exist.

significantly associated with their gender, age, and the presence of learning difficulties, whereas the parent-reported QoL was only associated with the presence of learning difficulties.

## Conclusions

Responses from both children and parents need to be considered to understand the similarities and differences between them and to provide further insight into the optimal way to help children with CHD effectively navigate the transition into adulthood. Future research studies of outcomes for survivors of children with CHD are needed to identify high-risk survivors for worse psychosocial functioning and assess prevention measures and treatment interventions to improve their QoL.

## Introduction

Congenital heart disease (CHD) comprises a wide variety of heart abnormalities, which together form the most common type of birth defects [1]. The incidence of CHD ranges between 4 and 15 per 1000 live births [2–4]. Hoffman and Kaplan suggest that the reasons behind the rather large difference in reported incidence rates are most likely due to variations in diagnostic accuracy and the criteria used for case inclusion; such that if even minor congenital heart defects were to be included then the incidence rates could potentially mount to 75 per 1000 births [5].

Mortality rates have significantly decreased for children with CHD. It is estimated that almost 85–90% of children with CHD manage to survive into adulthood [6,7]. However, the quality of life (QoL) of those children is often undermined due to their circulatory abnormalities and the medical and surgical therapies they undergo [8]. These factors may impact their physical, social, psychological, mental, and functional well-being. In fact, research has documented that the poor QoL of children with CHD may hinder healthy development [9]. Therefore, studying the QoL of this high-risk population is increasingly warranted. Available evidence on this matter is inconsistent. Some studies concluded that the QoL of children with CHD is poorer compared to normal children, which is a reasonable expectation [10]). Other studies have found that the QoL was not really compromised among children with CHD when compared to those without CHD [11,12]. The concept of QoL itself was also found to be perceived differently from the perspective of children, parents, and healthcare providers [13]. Other differential aspects related to QoL in children with CHD were reported in relation to their medical history. In a systematic review comprising 20 studies in high-income countries [14], the results revealed poorer QoL among children who underwent surgeries related to CHD than their healthy counterparts. Similar results were also reported in a study conducted in a lower middle-income country [15] where QoL was worse among CHD children with a positive correlation observed between the complexity of the disease and QoL impairment. Correlation of QoL with severity of the disease and multiple surgery was also concluded by a Swedish study [16]. However, a study specifically focused on children with repaired Tetralogy of Fallot found comparable QoL scores compared to healthy counterparts based on self-reporting. Interestingly, when parent-proxy reporting was considered, lower QoL scores were reported, with the physical aspect being the most affected [17].

In the Arab region, only two research studies on QoL of CHD children and their parents were conducted. The study by Azhar, AlShammasi, and Higgi in Saudi Arabia showed that the QoL of children and their parents were compromised at all levels [18]. As reported by the authors, findings were limited due to the use of a non-consensual questionnaire and the reliance on mother-reported assessments for their children. The other study, also from Saudi Arabia, assessed the QoL among parents of children with CHD but did not collect data from the children themselves. Findings revealed low QoL scores that are associated with the severity of the disease [19]. There is a need to contribute to the growing body of knowledge on QoL in congenital cardiology using data from under-researched populations. This is particularly relevant in the Arab region, where psychosocial problems are not considered to require professional help, and people who seek help often do not represent the broader community. Therefore, the purpose of this study was to evaluate QoL in children with CHD using an Arab sample from Jordan. The specific objectives were twofold: (1) to contrast the assessments of children's QoL reported by their parents with those reported by the children themselves, and (2) to assess the factors that influence the QoL of children with CHD. By incorporating the voices of both parents and children in QoL assessments, the validity and reliability of the results are enhanced.

## Methods

### Design

A cross-sectional design was implemented for this study, with data collection taking place at an education hospital in Jordan. This hospital was selected for being the first academic teaching hospital in Jordan, and also the first in the Arab region, the second in the Middle East, and the eighteenth worldwide to receive the accreditation of the Joint Commission International. It serves more than half a million patients a year. The hospital includes a pediatric cardiology unit, which has grown in its staff and capacity over the past decades to be one of the leading units in the region, accomplishing an excellent reputation in providing cardiac care for children, infants, and neonates [20]. All children aged between 2 and 18 years old with a confirmed diagnosis of CHD were eligible for the study. Exclusions were made for children with severe cognitive or physical impairments that could impede participation or hinder reliable data collection. Children whose parents declined their participation were also excluded. Parents of eligible children were invited to participate in the study during their follow-up visits through their primary cardiologist. Additionally, parents whose children met the eligibility criteria but did not have scheduled visits during the data collection period were contacted by their cardiologist via phone. The total number of parents who were contacted was 82. Of them, 79 consented to participate along with their children and returned the study questionnaire (response rate = 0.96). The study obtained IRB approvals from the University of Jordan and Jordan University Hospital.

The required sample size for this study was calculated from a power analysis based on multiple linear regression with eight predictors. The effect size was estimated based on prior published studies on QoL among children with CHD [18,19], which showed large effect sizes ($f^2$ = 0.30–0.35) for regression models. Accordingly, a two-sided alpha level was set at 0.05, with a power level of 0.8, and an effect size of $f^2$ = 0.3. The power analysis revealed that a minimum of 56 participants (children or parents) were needed for this study. Thus, 62 children-reported data and 79 parents-reported data in this study provided sufficient power for the proposed multiple regression analyses.

## Participants

A total of 79 children aged 2–18, all with a confirmed diagnosis of a CHD were included in the study along with their mothers. Of them, 30 (38.0%) were girls. About 67% were diagnosed with non-cyanotic CHD. Disease severity was determined based on the classification proposed by Hoffman and Kaplan, which categorizes CHD severity into three levels: mild, moderate, and severe [5]. Accordingly, 58.2% were categorized as having severe conditions, 40.5% moderate, while only 1.3% had mild conditions. Most of the children (92.4%) had undergone at least one operation, 81.0% had repaired defects, 13.9% underwent palliated procedures, and 24.1% were admitted to a neonatal intensive care unit after delivery. Table 1 details the sample characteristics.

## Measures

The study collected data on various sociodemographic characteristics pertaining to children. With CHD, including age group, gender, grade at school, primary diagnosis, cyanotic disease, cyanosis experience, disease severity, defect status, number of operations, learning difficulties, downgrading at school, admission to NICU, father's education, and mother's education. Diagnosis of the disease was established by a certified and licensed pediatric cardiologist.

The Pediatric Quality of Life Inventory (PedsQL, [20]) was used to assess QoL of children with CHD. The PedsQL includes 23 items based on a 5-point Likert scale (0 = not a problem;1 = almost never a problem; 2 = sometimes a problem; 3 = often a problem; 4 = almost always a problem). The inventory is available in different versions tailored for different age groups, including infants (ages 2–4), toddlers (ages 5–7), children (ages 8–12), and adolescents (ages 13–18). It consists of self-report questionnaires for older children and adolescents (ages 5–18), as well as proxy-report questionnaires completed by parents or caregivers for all children (ages 2–18). According to Varni, Seid, and Kurtin [21], The PedsQL can distinguish between healthy children and those with acute or chronic health conditions; it demonstrates good reliability and validity; and may be applicable in clinical trials, research, clinical practice, school health settings, and community populations. In this sample, almost all QoL scores had excellent values of Cronbach's alpha, except for a few reported by younger children (Table 2).

The PedsQL underwent a translation process in accordance with the translation guidelines established by the Mapi Research Trust). Initially, the questionnaire was translated from English to Arabic by one of the study investigators, who is proficient in both languages. Subsequently, a panel of five experts reviewed the translated questionnaire to assess its clarity and appropriateness in terms of translation. To ensure accuracy, a blind back-translation was performed by another bilingual expert, translating the Arabic version back into English. Additionally, pretesting and focus group discussions were conducted with a sample of 10 parents and their children representing the study population. The aim of this process was to confirm that the translated questionnaire was clear and suitable for use in the study.

## Data analysis

Descriptive statistics were first computed for describing the sample characteristics. The QoL scores were calculated based on the PedsQL measure's scoring manual [20]. Scores are transformed on a scale from 0 to 100 using two steps: First, items are reversed scored and linearly transformed to a 0–100 scale as follows: 0 = 100, 1 = 75, 2 = 50, 3 = 25, 4 = 0. Second, mean scores for subdimensions are calculated by summing the items divided by the number of items answered. The total QoL score equals the sum of all the items over the number of items answered on all the scales. ANOVAs were conducted for assessing the sensitivity of the difference in QoL scores among different age groups. Paired-sample $t$-tests were used to compare

**Table 1. Participants' characteristics (N = 79).**

| Characteristic | | *n* | % |
|---|---|---|---|
| **Age Group** *(for the age-appropriate versions of PedsQL used)* | 2–4 yrs | 17 | 21.5% |
| | 5–7 yrs | 18 | 22.8% |
| | 8–12 yrs | 24 | 30.4% |
| | 13–18 yrs | 20 | 25.3% |
| **Gender** | Female | 30 | 38.0% |
| | Male | 49 | 62.0% |
| **Grade at school** | No school | 1 | 1.3% |
| | Kindergarten | 7 | 8.9% |
| | 1st grade | 9 | 11.4% |
| | 2nd grade | 5 | 6.3% |
| | 3rd grade | 8 | 10.1% |
| | 4th grade | 4 | 5.1% |
| | 5th grade | 9 | 11.4% |
| | 6th grade | 3 | 3.8% |
| | 7th grade | 4 | 5.1% |
| | 8th grade | 6 | 7.6% |
| | 9th grade | 5 | 6.3% |
| | 10th grade | 4 | 5.1% |
| | 11th grade | 1 | 1.3% |
| | Unreported | 13 | 16.5% |
| **Primary diagnosis** | Patent ductus arteriosus | 3 | 3.8% |
| | Coarctation of the aorta (COA) | 8 | 10.1% |
| | Ventricular septal defect (VSD) | 17 | 21.5% |
| | Atrial septal defect (ASD) | 12 | 15.2% |
| | Tetralogy of Fallot (TOF) | 6 | 7.6% |
| | Ebstein anomaly | 1 | 1.3% |
| | Truncus arteriosus (TA) | 1 | 1.3% |
| | Transposition of great arteries (TGA) | 4 | 5.1% |
| | Total anomalous pulmonary venous return (TAPVR) | 1 | 1.3% |
| | Atrio-Ventricular Fistula (AV Fistula) | 1 | 1.3% |
| | Mixed (More than one heart disease) | 10 | 12.7% |
| | Single ventricle | 8 | 10.1% |
| | Aortic stenosis | 4 | 5.1% |
| | Pulmonary stenosis | 3 | 3.8% |
| **Cyanotic disease** | Yes | 26 | 32.9% |
| | No | 53 | 67.1% |
| **Cyanosis experience** | Never had cyanosis | 54 | 68.4% |
| | Had cyanosis only before being operated | 14 | 17.7% |
| | Chronic cyanosis | 11 | 13.9% |
| **Disease severity** | Mild | 1 | 1.3% |
| | Moderate | 32 | 40.5% |
| | Severe | 46 | 58.2% |
| **Defect status** | Repaired | 64 | 81.0% |
| | Not repaired | 4 | 5.1% |
| | Palliated | 11 | 13.9% |

*(Continued)*

**Table 1.** (Continued)

| Characteristic | | *n* | % |
|---|---|---|---|
| **Number of operations** | 0 | 6 | 7.6% |
| | 1 | 67 | 84.8% |
| | 2 | 3 | 3.8% |
| | 3 | 3 | 3.8% |
| **Learning difficulties** | Yes | 15 | 19.0% |
| | No | 38 | 48.1% |
| | Unreported | 26 | 32.9% |
| **Downgraded at school** | Yes | 8 | 10.1% |
| | No | 46 | 58.2% |
| | Unreported | 25 | 31.6% |
| **Admitted to NICU** | Yes | 19 | 24.1% |
| | No | 32 | 40.5% |
| | Unreported | 28 | 35.4% |
| **Father's education** | University (Masters, Bachelor's, Ph.D.) | 14 | 17.7% |
| | Diploma | 7 | 8.9% |
| | High school (Tawjeehi) | 10 | 12.7% |
| | School level | 17 | 21.5% |
| | Non (Illiterate) | 2 | 2.5% |
| | Unreported | 29 | 36.7% |
| **Mother's education** | University (Masters, Bachelor's, Ph.D.) | 9 | 11.4% |
| | Diploma | 8 | 10.1% |
| | High school (Tawjeehi) | 14 | 17.7% |
| | School level | 18 | 22.8% |
| | Non (Illiterate) | 1 | 1.3% |
| | Unreported | 29 | 36.7% |

QoL scores reported by children with those reported by their parents. Multiple linear regression was used to assess influencing factors of QoL of children with CHD.

Prior to conducting data analysis, the presence of missing or unreported data was diagnosed and treated. There were 3.8–36.7% of missing values on the study variables used in the analyses. Little's test for missing completely random (MCAR) was performed for assessing missing randomness. MCAR was assumed ($\chi^2(45) = 49.26$, $p = .307$), and no imputation for the missing values was necessary. Nevertheless, to preserve the sample size for achieving sufficient power, the missing values were imputed using the expectation-maximization algorithm. Except for descriptive statistics, all other statistical analyses were conducted using the imputed data. The missing data analysis and all the statistical analyses were implemented in IBM SPSS Statistics (Version 29.0) with the significance level set at .05.

## Results

### Aim one: Contrast levels of QoL from parents' perspectives to those of their children's perspectives

The study obtained self-reported (i.e., children-reported) scores on QoL from participants aged 5 to 18 years old. For children aged 2–4 years (*n* = 17), QoL scores were not applicable as they did not provide self-reported assessments. However, the QoL scores reported by parents of children in this age group which had a mean of 83.19 ± 14.57. Among children aged 5–7

**Table 2. Mean (M), standard deviation (SD), and Cronbach's alpha (α) of QoL scores by age group (N = 79).**

| Age Group | QoL Scores by Children | | | | QoL Scores by Parents | | | |
|---|---|---|---|---|---|---|---|---|
| | *n* | *M* | *SD* | *α* | *n* | *M* | *SD* | *α* |
| 2–4 yrs | Not applicable (n/a) | | | | 17 | 83.19 | 14.57 | .89 |
| 5–7 yrs | 18 | 91.78 | 4.43 | .46 | 18 | 79.45 | 17.83 | .93 |
| 8–12 yrs | 24 | 79.96 | 10.37 | .72 | 24 | 73.21 | 15.08 | .89 |
| 13–18 yrs | 20 | 79.43 | 16.09 | .91 | 20 | 71.32 | 19.64 | .95 |
| Total | 62 | 83.22 | 12.54 | n/a | 79 | 76.30 | 17.20 | n/a |
| ANOVA | F(2, 59) = 7.10, p < .002 | | | | F(3, 75) = 2.00, p = .121 | | | |

years, 18 participants provided self-reported QoL scores, which had a mean of 91.78 ± 4.43. In comparison, the corresponding scores reported by parents of children in this age group had a much lower mean of 79.45 ± 17.83. For the age group of 8–12 years, both children and parents reported lower QoL scores. The self-reported scores by 24 children had a mean of 79.96 ± 10.37, whereas the scores reported by their parents had a mean of 73.21 ± 15.08. Similarly, for children aged 13–18 years, both children and parents reported relatively lower QoL scores. The self-reported scores by 20 children had a mean of 79.43 ± 16.09, whereas the scores reported by their parents had a mean of 71.32 ± 19.64 (see Table 2).

Overall, when considering the total sample, including all age groups, the self-reported QoL scores by children had a mean of 83.22 ± 12.54. The corresponding scores reported by parents had a slightly lower mean of 76.30 ± 17.20. It is worth noting that the subgroup analysis for comparing mean QoL scores by age groups revealed that there were differences in children-reported QoL scores among age groups ($F(2, 59) = 7.10$, $p < .002$). However, there were no differences in parents-reported QoL scores among age groups ($F(3, 75) = 2.00$, $p = .121$) (see Table 2). In other words, QoL scores were sensitive to age-group differences reported by children but not by parents.

Paired-sample *t*-tests were conducted to contrast the assessments of children's QoL reported by their parents with those reported by the children themselves. The results revealed a divergence in perceptions of QoL between parents and children with parents reporting significantly lower scores. For children aged 5–7 years, the paired t-test showed a significant difference between the assessments reported by parents and children themselves (paired $t(17) = 3.08$, $p < .007$). Similarly, there was a significant difference between the two reports for children aged 8–12 years (paired $t(23) = 2.26$, $p < .033$), and those aged 13–18 (paired $t(19) = 2.73$, $p < .013$) (see Fig 1).

## Aim two: Factors influencing the QoL in children with CHD

Multiple linear regression analysis was conducted to examine influencing factors of children's QoL. Combined analysis for all ages was conducted to maximize predictive power. Besides age in years, the age group was also included in the model to account for the variation of QoL scores among age groups. In addition, the following variables were used as potential predictors: gender, disease type (cyanotic or not), disease severity, presence of learning difficulties, and parental education. Those predictors were selected based on the literature [22,23]. The regression analysis was conducted using two separate models: one for QoL scores reported by children ($N = 62$) and another for QoL scores reported by parents ($N = 79$).

The results revealed that the children model could significantly predict children reported QoL scores ($F(8, 53) = 4.96$, $p < .001$), explaining 42.8% of the variance in the outcome (QoL scores). The second model using parent-reported QoL scores was also significant ($F(8, 70) =$

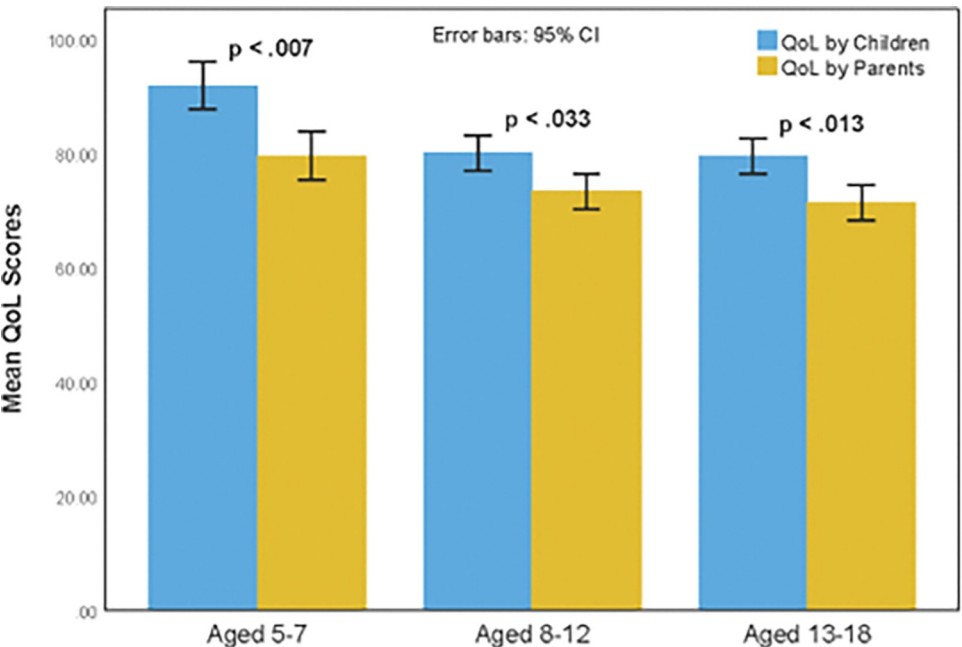

**Fig 1. Self-reported and Parent-reported QoL scores for children with CHD.**

3.31, $p < .003$) and could explain 27.4% of the variance in the outcome. Table 3 presents the estimated regression coefficients ($\beta$) with their standard errors ($SE$), $p$-values, and model fit statistics for the predictors of QoL scores reported by children and parents.

Considering the individual betas, the results revealed that the child's age was negatively associated with QoL scores, although the effect was significant only for children-reported scores ($p < .019$) but not for parents-reported scores ($p = .238$). Being female was associated with lower QoL scores. The effect was also significant for children ($p < .043$) but not for parents ($p = .183$). Neither the presence of cyanosis nor disease severity showed a significant association with QoL scores reported by either children or parents. Having learning difficulties was associated with lower QoL scores. The effect was significant for both children-reported scores ($p < .001$) and parents-reported scores ($p < .027$). Father's and mother's education

**Table 3. Estimated determinants of QoL from regression analysis.**

| Estimate | QoL Scores by Children (N = 62) | | | QoL Scores by Parents (N = 79) | | |
|---|---|---|---|---|---|---|
| | $\beta$ | $SE$ | $p$ | $\beta$ | $SE$ | $p$ |
| Intercept | 118.73 | 9.29 | < .001 | 77.58 | 10.39 | < .001 |
| Age group (*ordinal*) | -18.00 | 5.20 | .001 | -8.44 | 5.56 | .134 |
| Age (*yrs*) | -2.95 | 1.22 | .019 | -1.82 | 1.53 | .238 |
| Gender (*female vs. male*) | -6.26 | 3.02 | .043 | -5.18 | 3.85 | .183 |
| Cyanosis (*yes vs. no*) | -1.14 | 3.62 | .753 | -7.37 | 4.92 | .139 |
| Severe disease (*yes vs. no*) | -3.64 | 3.53 | .308 | 1.59 | 4.73 | .739 |
| Learning difficulties (*yes vs. no*) | -14.34 | 3.68 | < .001 | -10.51 | 4.66 | .027 |
| Father's education (*level*) | -.82 | 2.10 | .698 | 1.28 | 2.60 | .623 |
| Mother's education (*level*) | -1.14 | 1.83 | .537 | 3.11 | 2.43 | .205 |
| *Model fit (F, p; R²)* | *F(8, 53) = 4.96, p < .001; 42.8%* | | | *F(8, 70) = 3.31, p < .003; 27.4%* | | |

levels did not show significant associations with QoL scores reported by either children or parents.

## Discussion

Over the past few decades, there have been significant improvements in the management of CHDs, resulting in a higher survival rate for children with these conditions. While older studies on CHD outcomes have primarily focused on reporting mortality rates and treatment complications, mounting research efforts aimed at monitoring outcomes for every CHD case in many developed countries, with data being made publicly available [24]. However, studying QoL during childhood years has not paralleled the available literature on physical health outcomes [25]. Further, while there is emerging literature on QoL outcomes from developed countries, available literature from developing ones is scarce, and the Arab countries are no exception [18,19,26].

Cultural, financial, religious, and even geographical differences affect the perception and the meaning of QoL among different populations [27]. Therefore, studying QoL in different cultural backgrounds can provide important and unique data for every population. This study includes the first report on QoL in children with CHD residing in Jordan, with aims to compare the QoL assessments of children with CHD as reported by their parents and the children themselves, and to examine the factors influencing QoL in this specific population.

Overall, the results showed that younger children report significantly higher QoL scores. The preserved QoL in toddlers and young children with CHD can be attributed to several factors. Firstly, younger children may have limited awareness of their condition and its potential impact on their daily lives. They may be less likely to perceive and express the physical and emotional challenges associated with CHD, leading to higher reported QoL scores. Moreover, younger children may receive more focused attention and care from their parents and healthcare providers, which could contribute to better management of their condition and overall well-being. Parents of younger children may be more vigilant in ensuring medication adherence, regular medical check-ups, and creating a supportive environment, all of which can positively influence the child's QoL. On the other hand, older children reported relatively lower QoL scores, which is not surprising as these kids start to have their QoL unmasked and become more aware of their condition and its implications. They may experience physical limitations, such as decreased stamina or restricted activities, which can negatively impact their QoL [28]. Further, certain limitations or difficulties might become more apparent over time. In fact, neurodevelopmental impairments that affect children with CHDs may not be apparent until after the child is reading and performing more complex cognitive functioning [29]. Transitioning into adolescence is a critical period of development, and children with CHD may face unique challenges related to body image, self-esteem, and social interactions. They may experience emotional distress, anxiety, or feelings of being different from their peers, which can impact their overall well-being and QoL.

Several studies have addressed the divergence in QoL scores reported by children with CHD when compared to parents reported scores [30,31]. In this study, parents generally reported lower QoL scores for their children than what the children themselves reported. These findings come in line with previous literature. A systematic review [32] on QoL in children with CHD analyzed findings from 32 original articles and reported that the majority of clinical studies showed significant differences among children and their parents' responses regarding their QoL, with a tendency of children to report greater QoL scores than their parents. It can be argued that parents may have a more objective perspective on their child's QoL, taking into account the challenges and limitations imposed by the child's CHD. They

may be more aware of the medical interventions, treatment procedures, and potential long-term consequences of the condition, which could influence their perception of their child's well-being. From another perspective, parents may underestimate their child's ability imposing unnecessary restrictions that can in turn negatively impact their child's overall QoL and potential to grow and thrive. Such discrepancy is very important to be recognized as parents are often utilizers of the healthcare system. Understanding their perceptions of their child's disease can optimize the care for children with CHD.

The self-reported QoL for children in this study seemed to be significantly determined by a set of variables, including their gender, age, grade at school, disease type (cyanotic or not), disease severity, presence of learning difficulties, and parental education. It is recommended that healthcare providers take into consideration such factors in order to provide tailored interventions with better outcomes. It should be noted though that despite the strong association between this set of characteristics and QoL scores, the evidence is still conflicting on whether this association would be considered causal or not. Longitudinal investigations are warranted to better understand the temporality of such multifaceted relationships.

## Limitations

Reported responses might be limited due to the potential presence of selection bias. Parents who allowed their children to participate in the study may differ from those who did not, which could introduce a bias in the reported QoL scores. There may be underlying factors that influenced the decision of participation, such as the severity of the child's condition or the parents' perceptions of their child's well-being. Therefore, the findings may not fully reflect the QoL experiences of all children with CHD in the population. Furthermore, the study relied on self-reported measures of QoL, which may be subject to individual biases and variations. Children's and parents' perceptions of QoL may be influenced by various factors, such as their mood, current health status, or social desirability biases. Using collateral methods of assessment, such as clinician ratings along with children and parents reports, may provide a more comprehensive understanding of QoL in this population. Finally, the QoL scores reported by younger children had relatively low reliability. It is possible that this specific age group have different cognitive abilities, language skills, or developmental stages compared to older children, which could affect their responses and contribute to lower Cronbach's alpha scores. This suggests that measuring the construct of QoL among young children might need different assessment methods. Future studies could explore age-appropriate measures or adapt existing measures to better capture the QoL experiences of younger children.

## Conclusions

Future outcomes research studies in survivors of children with CHD should identify high-risk survivors for worse psychosocial functioning and assess prevention measures and treatment interventions to improve their QoL. This study provides evidence for the importance of integrating the QoL generic and cardiac modules as a component of the screening process for all children admitted with CHDs. This can provide relevant point-of-care feedback to allow for more comprehensive and proactive care. The holistic nursing approach is especially involved in achieving this goal. Responses of both children and parents need to be considered to understand the similarities and differences between them and to provide further insight into the optimal way to assess QoL. Understanding the effects of parent's perceptions of their child's QoL in a child with a CHD is important and necessary to evaluate outcomes of care, better plan care coordination around the family's needs, and inform policies around family-centered care and care for children with CHDs. It is also crucial to have QoL assessment tools that are

feasible, meaningful, and clinically useful to identify and proactively intervene with patients who are at high risk in terms of neurodevelopmental, psychosocial, and physical dysfunction. Such an approach would help children with CHD effectively navigate the transition into adulthood.

## Author Contributions

**Conceptualization:** Latefa Ali Dardas, Ahmad Eid Ashakhanba, Iyad Al-Ammouri.

**Data curation:** Latefa Ali Dardas, Wei Pan, Mohammad Naim Abid.

**Formal analysis:** Latefa Ali Dardas, Wei Pan.

**Funding acquisition:** Latefa Ali Dardas.

**Investigation:** Ahmad Imad Hamdan, Raghed Abdel Hay Abu Jabeh, Omar Sami Abdelhai, Mohammad Naim Abid, Hashim Ahmad Mohammad, Iyad Al-Ammouri.

**Methodology:** Latefa Ali Dardas, Ahmad Imad Hamdan, Raghed Abdel Hay Abu Jabeh, Ahmad Eid Ashakhanba, Omar Sami Abdelhai, Mohammad Naim Abid, Hashim Ahmad Mohammad, Iyad Al-Ammouri.

**Project administration:** Latefa Ali Dardas.

**Supervision:** Latefa Ali Dardas, Iyad Al-Ammouri.

**Validation:** Latefa Ali Dardas.

**Visualization:** Wei Pan.

**Writing – original draft:** Latefa Ali Dardas, Wei Pan, Ahmad Imad Hamdan, Raghed Abdel Hay Abu Jabeh, Ahmad Eid Ashakhanba, Omar Sami Abdelhai, Mohammad Naim Abid, Hashim Ahmad Mohammad, Iyad Al-Ammouri.

**Writing – review & editing:** Latefa Ali Dardas, Wei Pan, Ahmad Imad Hamdan, Raghed Abdel Hay Abu Jabeh, Ahmad Eid Ashakhanba, Omar Sami Abdelhai, Mohammad Naim Abid, Hashim Ahmad Mohammad, Iyad Al-Ammouri.

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
