## [Decision Letter · Decision Letter 0]

7 May 2023

PONE-D-23-03793Quality of Life in Arab Children and Adolescents with Congenital Heart DiseasePLOS ONE

Dear Dr. Dardas,

Thank you for submitting your manuscript to PLOS ONE. After careful consideration, we feel that it has merit but does not fully meet PLOS ONE’s publication criteria as it currently stands. Therefore, we invite you to submit a revised version of the manuscript that addresses the points raised during the review process.

I have reviewed the reviewers' comments, and personally see that  we should commend the authors for the idea and efforts spent to implement this manuscript. Please respond as much as possible to the raised comments to expedite its publication. ==============================

We look forward to receiving your revised manuscript.

Kind regards,

Antoine Fakhry AbdelMassih

Academic Editor

PLOS ONE

Journal Requirements:

Reviewers' comments:

Reviewer's Responses to Questions

**Comments to the Author**

1. Is the manuscript technically sound, and do the data support the conclusions?

Reviewer #1: Yes

Reviewer #2: Partly

Reviewer #3: Partly

Reviewer #4: Yes

2. Has the statistical analysis been performed appropriately and rigorously? 

Reviewer #1: Yes

Reviewer #2: No

Reviewer #3: No

Reviewer #4: Yes

3. Have the authors made all data underlying the findings in their manuscript fully available?

Reviewer #1: Yes

Reviewer #2: No

Reviewer #3: No

Reviewer #4: Yes

4. Is the manuscript presented in an intelligible fashion and written in standard English?

Reviewer #1: Yes

Reviewer #2: No

Reviewer #3: No

Reviewer #4: No

5. Review Comments to the Author

Reviewer #1: Review comments on Manuscript Number: PONE-D-23-03793. Entitled " Quality of Life in Arab Children and Adolescents with Congenital Heart Disease"

Overall, the idea of research is very interesting to be studied nowadays and paper is coherently developed. However, there are some comments and suggestions.

Title

- Well structured

Abstract

- Well structured

Introduction

- Well structured

Materials and methods

- The participants sections is recommended to be moved after study design

Statistical analysis

- Well structured

- The subtitles (Aim One: Compare the levels of QoL from parents’ perspectives to those of their children’s perspectives/ Aim Two: assess determinants of QoL in children and adolescents with CHD). You may replace by (levels of QoL from parents’ perspectives to those of their children’s perspectives/ determinants of QoL in children and adolescents with CHD)

Discussion

- Well structured

Reviewer #2: Thank you so much for your efforts

1-some sentences need to be revised ,e.g. the first sentence in the abstract . also in the method section {Parents of children who were eligible but did not have arranged visits for their children during the data collection period were contacted over phone by their cardiologist} as their cardiologist could be misleading for parents not children.

2-Regarding the title ,it is better to add in Jordanian children instead of Arab

3-In abstract : result section authors wrote [Social functioning domain showed the highest scores, while school

functioning domain showed the lowest scores] to attract reader add these scores

4-in design : authors wrote [and also the first in the Arab region, the second in the Middle East,

and the eighteenth worldwide to receive the accreditation of the Joint Commission International.] need to add reference or source

5- Add clear inclusion or eligibility criteria

6-Can we say majority for 67% ?

7- The justification for classifying children to :13-18/ 8-12 etc is not clear .Also need to add the number of age group in table 1 and clarify the effect of age on the results.

8- what do you mean by parents , father and mother , as I saw only father level of education in table 1. please specify ?

9- Did you collect data regarding level of social standard

10-The results for the second aim need to be more clear , add in table instead of just text and shows how these number/values for F of R came from . i could not find effect of gender, grade at school, disease type (cyanotic or not), disease severity (mild, moderate, severe), presence of learning difficulties, and parental education. Data should clarify if gender affect quality of life and in what aspect , also what was affected by the severity of illness .

11-Did the parents education affect the quality of life

12- regarding table 1 : what do you mean by missing ?

13-Also in table 1 : total number in raw of level Father education is 50 ? please verify / Also change the title of the table to show that include the children (79) and the fathers (50) .

14-For all tables add all abbreviations as Q1 ,Q3 and name of test below the tables .

15-whay in age 8-12 number of children 21 and parents 23 ?Also in age 13-18 , children 19 and parents 18

16- when I calculate the number of parents in table 2 it was 76 ? please explain the discrepancy , in table 1 50 and table 2 it is 76 ?

17- Why no results for the Cronbach’s Alpha test

18-please elaborate from results of the study on what is written in the conclusion [This study shows targeting physical domains of care are important for younger children,]

Reviewer #3: 1- The introduction section is very long and clumsy, better to re-written in a more specified way.

2- There is no mention of sample size calculation anywhere in the manuscript! This needs to be clearly stated.

3- Did you use an Arabic translated version of the "The Pediatric Quality of Life Inventory"? How was it validated? Did you get permission from the WHO? who administered this?

4- The statistical analysis needs to be strengthened. There is a need for re-analyzing it with emphasis on subgroup comparisons, if dose response relationship of severity of CHD with HRQOL is to be examined.

5- The authors can also assess the sensitivity of the instrument used by testing the difference in scores between different groups.

6- The discussion is vague and does not focus on the exact results from the study. There is no proper order of the sections within the discussion. This section needs to be restructured - state the major findings first, the meaning of those findings next, followed by relating individual findings with other published literature. Give emphasis on clinical relevance, followed by strengths, limitations, and message to future researchers if any.

7- The discussion section needs to be re-written in a structured way.

8- The reference section needs to be revised according to the PLOS ONE authors guidelines.

9- The English language should be checked by a Native English speaker.

Reviewer #4: the authors performed a cross-section study about QoL in children with CHD. the topic is important and interesting. the manuscript is organized and the methodology is sound but i have few comments that could improve the manuscript:

1- the manuscript needs good English editing

2-introduction is too long and better to be shortened.

6. PLOS authors have the option to publish the peer review history of their article (what does this mean?). If published, this will include your full peer review and any attached files.

Reviewer #1: No

Reviewer #2: **Yes: **Safaa ELMeneza

Reviewer #3: No

Reviewer #4: **Yes: **Doaa El Amrousy

---

## [Author Response · Author response to Decision Letter 0]

7 Jul 2023

Journal: PLOS One

Manuscript ID: PONE-D-23-03793

Title: Quality of Life in Arab Children with Congenital Heart Disease

Dear Editor;

Thank you for the opportunity to revise and resubmit our manuscript (PONE-D-23-03793). We appreciate the valuable comments from the reviewers. All of the comments were carefully addressed as outlined below. We believe the manuscript is stronger as a result of these edits and look forward to hearing from you. 

Author's Reply to the Review Report (Reviewer 1)

Reviewer #1: Review comments on Manuscript Number: PONE-D-23-03793. Entitled " Quality of Life in Arab Children and Adolescents with Congenital Heart Disease"

Overall, the idea of research is very interesting to be studied nowadays and paper is coherently developed. However, there are some comments and suggestions.

Title: Well structured

Abstract: Well structured

Introduction: Well structured

Materials and methods: The participants sections is recommended to be moved after study design

Statistical analysis

- Well structured

- The subtitles (Aim One: Compare the levels of QoL from parents’ perspectives to those of their children’s perspectives/ Aim Two: assess determinants of QoL in children and adolescents with CHD). You may replace by (levels of QoL from parents’ perspectives to those of their children’s perspectives/ determinants of QoL in children and adolescents with CHD)

Discussion

- Well structure

RESPONSE: Thank you so much for the positive feedback. We hope that our revisions have further strengthened the work. We also addressed the suggested helpful edits on aims subtitles. 

Author's Reply to the Review Report (Reviewer 2)

Reviewer #2: Thank you so much for your efforts

1-some sentences need to be revised ,e.g. the first sentence in the abstract . also in the method section {Parents of children who were eligible but did not have arranged visits for their children during the data collection period were contacted over phone by their cardiologist} as their cardiologist could be misleading for parents not children.

RESPONSE: Thank you so much for the positive feedback. We hope that our revisions have further strengthened the work. We also addressed the suggested helpful changes to edit the noted sentences. The entire manuscript was also indeed critically revised by a native English speaker.

2-Regarding the title ,it is better to add in Jordanian children instead of Arab.

RESPONSE: Thank you for this suggestion, However, some of our participants were not Jordanians (i.e., Syrians and Palestinians). Thus, we chose the broader term Arabs.

4-in design : authors wrote [and also the first in the Arab region, the second in the Middle East,

and the eighteenth worldwide to receive the accreditation of the Joint Commission International.] need to add reference or source

RESPONSE: Thank you for noting this. Reference was added.

5- Add clear inclusion or eligibility criteria

RESPONSE: Thank you for noting this. Clear inclusion or eligibility criteria were added under the design section.

6-Can we say majority for 67% ?

RESPONSE: Got it. We edited the language throughout the manuscript to ensure objective description of the sample.

7- The justification for classifying children to :13-18/ 8-12 etc is not clear .Also need to add the number of age group in table 1 and clarify the effect of age on the results.

RESPONSE: Agreed we did not clearly describe this. In fact, it was not our classification. This is based on the tool used for measuring QOL. The Pediatric Quality of Life Inventory (PedsQL) which is often used to assess QoL of children and adolescents with CHD has several versions based on the four age groups used. We added clarification on the measure and also edited the tables as suggested to better clarify age groups.

8- what do you mean by parents , father and mother , as I saw only father level of education in table 1. please specify?

RESPONSE: Yes, we meant both fathers and mothers. Table 1 includes both results. Not sure why the last cell didn’t appear in the pdf. We made sure its present in the revised version. 

9- Did you collect data regarding level of social standard.

RESPONSE: Unfortunately, not.

10-The results for the second aim need to be more clear , add in table instead of just text and shows how these number/values for F of R came from . i could not find effect of gender, grade at school, disease type (cyanotic or not), disease severity (mild, moderate, severe), presence of learning difficulties, and parental education. Data should clarify if gender affect quality of life and in what aspect , also what was affected by the severity of illness .

RESPONSE: Thank you. Yes, we agree that adding a table would make the results much more clear. We then clarified in detail the effect of the individual betas in the model.

11-Did the parents education affect the quality of life.

RESPONSE: We clarified in the text that father's and mother's education levels did not show significant associations with QoL scores reported by either children or parents.

12- regarding table 1 : what do you mean by missing ? 

RESPONSE: It means unreported. We used the term ‘unreported’ instead of missing to avoid confusion. 

13-Also in table 1 : total number in raw of level Father education is 50 ? please verify / Also change the title of the table to show that include the children (79) and the fathers (50) .

RESPONSE: Thank you for noting this. We agree that the cells are not clear. We edited the table titles, cells, and format to better clarify content. As per the discrepancy in total numbers, it was because some participants did not respond to these questions. We labeled those as ‘unreported’.

14-For all tables add all abbreviations as Q1 ,Q3 and name of test below the tables .

RESPONSE: We indeed formatted table content to include the means and standard deviations only to make them clearer. The entire dataset is associated with the manuscript for readers interested in further statistical details.

15-whay in age 8-12 number of children 21 and parents 23 ?Also in age 13-18 , children 19 and parents 18?

RESPONSE: Thank you for noting this. We reformatted table content to avoid the confusion when comparing the content of each cell. Now table 2 includes parallel content for children and parent versions of the scale so the reader can easily compare the number of participants across different age groups. The entire dataset is associated with the manuscript for readers interested in further statistical details.

16- when I calculate the number of parents in table 2 it was 76 ? please explain the discrepancy , in table 1 50 and table 2 it is 76 ?

RESPONSE: Thank you for noting this. We reformatted the table content to avoid confusion when comparing the content of each cell. Now table 2 includes parallel content for children and parent versions of the scale so the reader can easily compare the number of participants across different age groups. The entire dataset is associated with the manuscript for readers interested in further statistical details.

17- Why no results for the Cronbach’s Alpha test

RESPONSE: Thank you for noting this. We added Cronbach’s alpha results in table 2 and also discussed their results.

18-please elaborate from results of the study on what is written in the conclusion.

RESPONSE: Thank you for noting this. We expanded the results section as well as the discussion sections to tap all aspects of the study.

Author's Reply to the Review Report (Reviewer 3)

Reviewer #3: 1- The introduction section is very long and clumsy, better to re-written in a more specified way.

RESPONSE: Thank you for noting this. The entire introduction was re-written and organized as suggested. 

2- There is no mention of sample size calculation anywhere in the manuscript! This needs to be clearly stated.

RESPONSE: Thank you for noting this. We clarified in text that the sample size was calculated based on the following parameters:

The required sample size for this study was calculated based on multiple linear regression with eight predictors. The effect size was estimated based on prior published studies on QOL among children with CHD, which showed large effect sizes (f2 = 0.30-0.35) for the prediction model used in this study. Accordingly, a two-sided alpha level was set at 0.05, with a power level of 0.8, and an effect size of f2 = 0.3. The power analysis revealed that a minimum of 56 participants (children or parents) were needed for this study. Thus, 62 children-reported data and 79 parents-reported data in this study provided sufficient power for the proposed multiple regression analyses.

3- Did you use an Arabic translated version of the "The Pediatric Quality of Life Inventory"? How was it validated? Did you get permission from the WHO? who administered this?

RESPONSE: Thank you for highlighting these issues. We indeed conducted tremendous efforts to translate the scale into Arabic. Details regarding the translation process were added to the manuscript under the measures section. As per the permission, this scale is not from the WHO. Permission was granted from the copyright owner (details can be found here https://eprovide.mapi-trust.org/instruments/pediatric-quality-of-life-inventory )

4- The statistical analysis needs to be strengthened. There is a need for re-analyzing it with emphasis on subgroup comparisons, if dose response relationship of severity of CHD with HRQOL is to be examined.

RESPONSE: Thank you for this suggestion, which had really given us new insights into how to present the data. The analysis was robustly strengthened along with the reporting style and supporting tables. 

5- The authors can also assess the sensitivity of the instrument used by testing the difference in scores between different groups.

RESPONSE: Thanks for the suggestion. The sensitivity of the difference in scores among different groups was assessed by ANOVAs whose results are reported in the results section.

6- The discussion is vague and does not focus on the exact results from the study. There is no proper order of the sections within the discussion. This section needs to be restructured - state the major findings first, the meaning of those findings next, followed by relating individual findings with other published literature. Give emphasis on clinical relevance, followed by strengths, limitations, and message to future researchers if any. 7- The discussion section needs to be re-written in a structured way.

RESPONSE: Thank you for this suggestion. As mentioned earlier, the entire discussion section was rewritten and organized focusing on the results and moving forward to future recommendations. 

8- The reference section needs to be revised according to the PLOS ONE authors guidelines.

RESPONSE: Done! References in the text and in the list were edited according to the journal’s style. 

9- The English language should be checked by a Native English speaker.

RESPONSE: Agreed! The entire manuscript was critically revised by a native English speaker.

Author's Reply to the Review Report (Reviewer 4)

Reviewer #4: the authors performed a cross-section study about QoL in children with CHD. the topic is important and interesting. the manuscript is organized and the methodology is sound but i have few comments that could improve the manuscript:

1- the manuscript needs good English editing

2-introduction is too long and better to be shortened.

RESPONSE: Thank you so much for the positive feedback. We hope that our revisions have further strengthened the work. The introduction was rewritten in a more concise way, and the entire manuscript was critically revised by a native English speaker.

Thank you!

---

## [Decision Letter · Decision Letter 1]

7 Aug 2023

Quality of Life in Arab Children with Congenital Heart Disease

PONE-D-23-03793R1

Dear Dr. Dardas,

We’re pleased to inform you that your manuscript has been judged scientifically suitable for publication and will be formally accepted for publication once it meets all outstanding technical requirements.

Kind regards,

Antoine Fakhry AbdelMassih

Academic Editor

PLOS ONE

Additional Editor Comments (optional):

Reviewers' comments:

Reviewer's Responses to Questions

**Comments to the Author**

1. If the authors have adequately addressed your comments raised in a previous round of review and you feel that this manuscript is now acceptable for publication, you may indicate that here to bypass the “Comments to the Author” section, enter your conflict of interest statement in the “Confidential to Editor” section, and submit your "Accept" recommendation.

Reviewer #1: All comments have been addressed

Reviewer #2: All comments have been addressed

Reviewer #3: (No Response)

Reviewer #4: All comments have been addressed

2. Is the manuscript technically sound, and do the data support the conclusions?

Reviewer #1: Yes

Reviewer #2: Yes

Reviewer #3: Partly

Reviewer #4: Yes

3. Has the statistical analysis been performed appropriately and rigorously? 

Reviewer #1: Yes

Reviewer #2: Yes

Reviewer #3: Yes

Reviewer #4: Yes

4. Have the authors made all data underlying the findings in their manuscript fully available?

Reviewer #1: Yes

Reviewer #2: Yes

Reviewer #3: No

Reviewer #4: Yes

5. Is the manuscript presented in an intelligible fashion and written in standard English?

Reviewer #1: Yes

Reviewer #2: Yes

Reviewer #3: Yes

Reviewer #4: Yes

6. Review Comments to the Author

Reviewer #1: Review comments on Manuscript Number: (PONE-D-23-03793R1) entitled ‘’ Quality of Life in Arab Children with Congenital Heart Disease''.

Overall, this study provides a novel approach. The idea of research is very interesting, well written and reasonable. I would like to thank the authors for their successful work to address the reviewers' comments. The authors have done great efforts to accomplish this work. They fulfilled all comments and made necessary changes throughput the manuscript. I recommend accepting the manuscript its revised form.

Reviewer #2: Thank you for response to the comments and the required changes as well as the modifications that were suggested .

Reviewer #3: The manuscript was much improved. But still there are some vital issues need to be explained:

- As i mentioned before in point (3) of my previous revision regarding "the use of the Arabic translated version of the "The Pediatric Quality of Life Inventory"? How was it validated? Did you get permission from the WHO? who administered this?", the authors clarified all the details about translation and validation and added this to the manuscript which added much strength to the manuscript, BUT when returning to the site they provide (https://eprovide.mapi-trust.org/instruments/pediatric-quality-of-life-inventory): it denoted that any author who needs a translated version MUST submit his request for translation online and the Inventory Owners will PROVIDE team will send the authors a translation agreement along with Linguistic Validation Guidelines. Did the authors made all these steps?? If yes, please provide the journal a copy of the "translation agreement".

Also, don't write in the measure section "The PedsQL underwent a translation process in accordance with the translation guidelines established by the World Health Organization (WHO, 2005)" !!!!!!! better if you have the "Linguistic Validation Guidelines" from the Mapi Research Trust, so re-write it as "The PedsQL underwent a translation process in accordance with the translation guidelines established by the Mapi Research Trust"

-- As i mentioned before in point (8) of my previous revision regarding "The reference section needs to be revised according to the PLOS ONE authors guidelines", but a quick revision i noticed that the references must be checked again; for example: many references are missing there full author list or even the "et al." i.e. reference no. 19, also sthe style of references are not the same i.e. reference no. 20. also there are two references carrying the no. (20) and this interrupted the logic sequence of references!!!!!!!

---As the language of the manuscript is improved, BUT more professional English language editing service is needed.

Reviewer #4: The authors have adequately addressed all my comments. The methodology is technically sound the data support the conclusion. the statistical analysis has been performed appropriately and rigorously. The manuscript presented in an intelligible fashion and written in standard English.

7. PLOS authors have the option to publish the peer review history of their article (what does this mean?). If published, this will include your full peer review and any attached files.

Reviewer #1: No

Reviewer #2: **Yes: **safaa ELMeneza

Reviewer #3: No

Reviewer #4: **Yes: **Doaa El Amrousy

---

## [Editor Report · Acceptance letter]

7 Sep 2023

PONE-D-23-03793R1 

Quality of Life in Arab Children with Congenital Heart Disease 

Dear Dr. Dardas:

I'm pleased to inform you that your manuscript has been deemed suitable for publication in PLOS ONE. Congratulations! Your manuscript is now with our production department. 

Kind regards, 

on behalf of

Prof Antoine Fakhry AbdelMassih 

Academic Editor

PLOS ONE